# Learning Generalizable Manipulation Policies with Object-Centric 3D Representations

**Yifeng Zhu**[1], **Zhenyu Jiang**[1], **Peter Stone**[1,2], **Yuke Zhu**[1]

[1]The University of Texas at Austin [2]Sony AI

**Abstract:** We introduce GROOT, an imitation learning method for learning robust policies with object-centric and 3D priors. GROOT builds policies that generalize beyond their initial training conditions for vision-based manipulation. It constructs object-centric 3D representations that are robust toward background changes and camera views and reason over these representations using a transformer-based policy. Furthermore, we introduce a segmentation correspondence model that allows policies to generalize to new objects at test time. Through comprehensive experiments, we validate the robustness of GROOT policies against perceptual variations in simulated and real-world environments. GROOT's performance excels in generalization over background changes, camera viewpoint shifts, and the presence of new object instances, whereas both state-of-the-art end-to-end learning methods and object proposal-based approaches fall short. We also extensively evaluate GROOT policies on real robots, where we demonstrate the efficacy under very wild changes in setup. More videos and model details can be found in the appendix and the project website https://ut-austin-rpl.github.io/GROOT.

**Keywords:** Manipulation, Object-Centric Representation, Imitation Learning

## 1  Introduction

Building robots capable of performing diverse manipulation tasks necessitates a rich repertoire of sensorimotor skills. These skills must operate on real-world perception and generalize across changing environments. Imitation learning (IL) [1–8] has recently emerged as a practical and efficient approach for training neural network-based visuomotor policies to tackle complex manipulation tasks. However, existing IL methods face difficulties in generalizing beyond the training environments from which the demonstration data are collected. Consequently, applying these methods to real-world tasks would entail time-consuming data collection and model re-training for each setting.

To harness the strengths of IL algorithms in learning visuomotor policies for real-world tasks, enhancing their robustness and adaptability to environmental variations becomes crucial. Prior work [9–13] has suggested that end-to-end learning policies are particularly brittle to even slight visual variations in the environment, including changes in the background, shifts in camera viewpoints, and presence of new object instances. As a result, IL policies trained on a limited set of demonstrations are commonly evaluated in the same workspace with strictly controlled conditions, with fixed backgrounds and meticulously calibrated cameras.

In this work, we argue that incorporating **structured visual representations** into the design of IL policies is important for enhancing their capability to handle visual variations naturally encountered in real-world deployments. In particular, a desirable visual representation should possess the following two properties: **object-centric** and **3D-aware**. The object-centric property exploits the compositional structure of visual scenes as a set of objects and entities. It encourages the policies to focus on task-relevant factors while minimizing the impacts of visual distractors. The 3D-aware property lifts the spatial reasoning from the 2D plane to a unified reference frame of 3D coordinates. It improves the representation's spatial invariance against changes in camera viewpoints.

7th Conference on Robot Learning (CoRL 2023), Atlanta, USA.

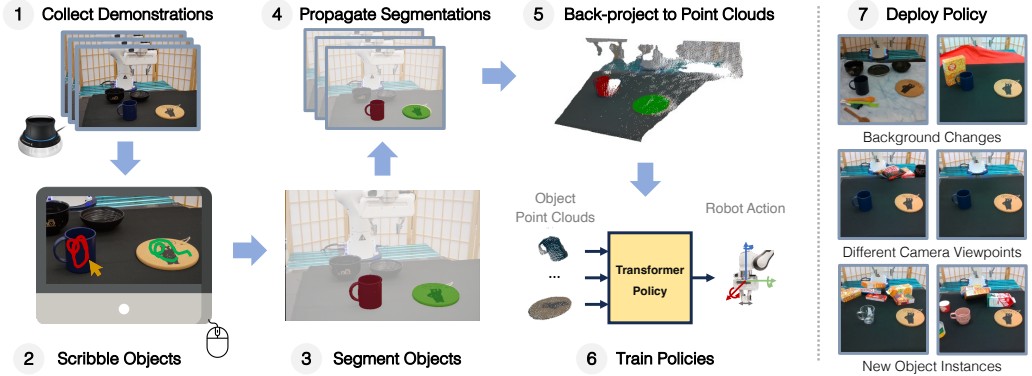

Figure 1: **GROOT Overview.** GROOT learns closed-loop visuomotor policies from demonstrations under a single setup, and generalizes to setups with different conditions, namely different visual distractions, changed camera angles, and new objects.

We present GROOT (**G**eneralizable **RO**bot Manipulation Policies for Visu**O**motor Con**T**rol), which aims to learn visuomotor policies that generalize beyond the conditions where demonstrations are collected. Fig. 1 provides an overview of our approach. Central to GROOT is building object-centric 3D representations that exhibit robustness against visual variations. We harness the power of large-scale, open-vocabulary visual recognition models newly developed by the vision community [14–18]. Building the representations requires three major steps. First, GROOT uses an interactive segmentation model [19] that allows demonstrators to annotate instance segmentation masks of *task-relevant* objects by simply scribbling on *a single frame*. Task-relevant objects here refer to foreground objects involved in a given task. Second, GROOT propagates the segmentation masks to the rest of the demonstration frames using a pretrained Video Object Segmentation (VOS) model [17]. Finally, GROOT back-projects 2D segmentation masks into 3D point clouds using depth from the RGB-D observations [20]. These point clouds are then transformed into a predetermined reference frame invariant to the camera viewpoints, such as the robot base frame. The segmented point clouds capture object information like geometry and their spatial relations in 3D space. Finally, the depth point clouds of individual objects are encoded into a discrete set of tokens, which are further processed by a transformer-based policy [3, 21] to predict actions. We train our encoders and policies end-to-end with behavior cloning objectives.

To generalize to new object instances during deployment, we introduce a segmentation correspondence model. This model utilizes an open-vocabulary segmentation model [14] to generate a set of object masks from visual observations of deployment environments. For each annotated object mask from the training phase, the model identifies its closest match based on DINOv2 [16] semantic feature similarities. The model can, therefore, propagate the object masks from training environments to test scenes, enabling GROOT to apply the learned policies to unseen objects from the same categories. For example, a policy trained to pick up one mug can be reused to pick up new mugs.

We conduct comprehensive experiments in both simulation and the real world. In simulation, results show that our learned policies excel at handling all generalization dimensions, while baselines based on end-to-end learning or object proposal priors fail at large variations in perception. In five real-robot tasks, we demonstrate that GROOT can generalize strongly to diverse backgrounds, different camera angles, and new object instances.

## 2 Related Work

**Deep Imitation Learning for Robot Manipulation** Deep imitation learning (IL) has emerged as a highly efficient method for learning end-to-end manipulation policies in the real world [1, 6, 8, 22–25]. New IL algorithms have also shown promise in handling long-horizon and multi-stage manipulation tasks [3, 26]. However, most of these approaches assume the test environments closely resemble the development environments in which the demonstrations are collected. This assumption hampers learned policies from being deployed beyond their training conditions. To improve the

broader applicability of IL methods, GROOT focuses on training policies using data from a single environment and enabling them to generalize across various visual variations, including background changes, different camera viewpoints, and new object instances.

**Representations Learning for Vision-Based Manipulation**   Various types of visual representations have been studied for vision-based manipulation policies. Early work commonly uses intermediate visual representations of known objects like bounding boxes [27–29], making it hard to generalize to new object instances. End-to-end learning methods seek to directly map sensory inputs to actions with neural networks [23, 30], but they are susceptible to covariate shifts and causal confusion [31]. Much literature has investigated inductive biases and learning techniques to overcome issues of end-to-end learning: pretraining on large dataset [32, 33], generative augmentation [34–36], spatial attention [37, 24], and affordances [25, 38]. However, these inductive biases are either tailored to well-defined motion primitives or require finetuning but yet have shown limited performance in downstream tasks [39]. Recent work has also shown great efficacy in leveraging task-agnostic object-centric priors to solve challenging manipulation tasks [3, 40], but the policy performance are restricted within the training setting. Our work also uses object-centric priors, but unlike prior works, our approach is able to learn robust policies that generalize beyond the training settings.

**Object-Centric Representation Learning**   Robotics and vision communities have extensively explored using object-centric representations to reason about visual scenes in a modular way. In robotics, researchers commonly use poses [41–43] and bounding boxes [44, 45] to represent objects presented in a scene. These representations are confined to known object categories or instances. Recent work leverages unsupervised learning approaches [46, 47] to endow the manipulation policies with object awareness [48, 49], but these approaches are limited to simulation domains and fall short in generalizing to real-world tasks. Recent advances in open-world visual recognition have produced large models, including image segmentation [14], video object segmentation [17], and semantic features [15, 16]. These models show great generalization abilities that can perceive objects beyond certain categories, making them suitable for robot manipulation tasks, which require robots to perceive and understand a large variety of objects. GROOT leverages these pretrained models to construct effective object-centric 3D representations for more generalizable policies than prior works.

## 3   GROOT

We present GROOT, an imitation learning method for learning generalizable policies of vision-based manipulation. Our core idea is to factorize raw RGB-D images observed from a calibrated camera into segmented point clouds of task-relevant objects, forming object-centric 3D representations for policy learning. Fig. 2 illustrates our method. We first formulate the problem of learning vision-based manipulation policies and define the dimensions of generalization we focus on. We then introduce how we build our object-centric 3D representations. Following the representations, we describe our transformer-based policy. Finally, we introduce the segmentation correspondence model that allows our policies to generalize to new object instances.

### 3.1   Problem Formulation

We formulate a vision-based manipulation task as a finite-horizon Markov Decision Process (MDP), which is described by a tuple $< \mathcal{S}, \mathcal{A}, \mathcal{P}, T, \mu, R >$, where $\mathcal{S}$ is the state space of raw sensory data including RGB-D images and robot proprioception, $\mathcal{A}$ is the action space of low-level motor commands (typically 20Hz OSC commands), $\mathcal{P}$ is the transition dynamics $\mathcal{S} \times \mathcal{A} \mapsto \mathcal{S}$, $T$ is the maximal horizon of a task, $\mu$ is the initial state distributions of a task, and $R$ is the sparse reward function that returns reward +1 when the task goals are satisfied. This work considers task goals with clear semantic interpretation, such as "a mug is placed on the coaster." Such task definitions allow us to clearly define the goals for manipulating unseen objects from the same category. The ultimate goal is to find a visuomotor policy $\pi$ that maximizes the expected task success rates.

In this paper, we assume that we are given $N$ demonstration trajectories $D = \{\tau_i\}_{i=1}^N$ collected through teleoperation. Each trajectory $\tau_i$ is a sequence of state-action pairs $(s_1, a_1, \ldots, a_{T-1}, s_T)$. Our method uses the Behavioral Cloning (BC) algorithm to learn $\pi$ from $D$. BC learns a policy that

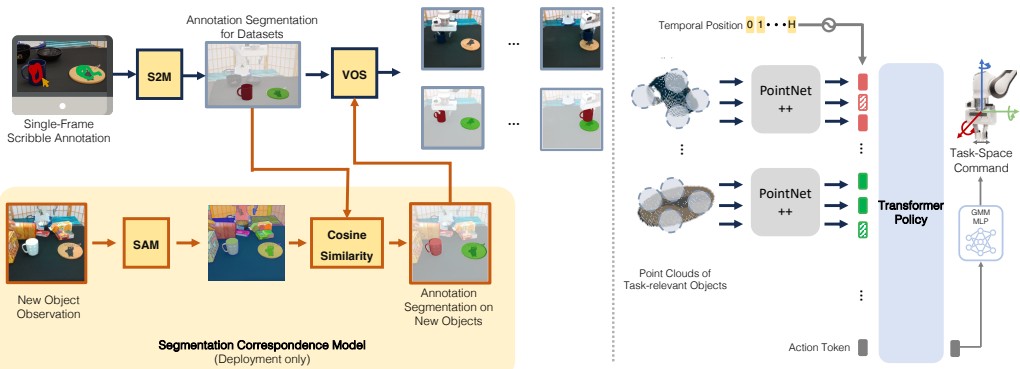

Figure 2: **GROOT Model Architecture.** GROOT leverages an interactive segmentation model, S2M, to obtain a single-frame annotation from demonstrators. Then a Video Object Segmentation model, XMem, propagates segmentation masks across time frames. The object masks are then back-projected into point clouds, and a transformer-based policy processes the point clouds to output actions. During deployment, GROOT uses a segmentation correspondence model based on an open-vocabulary segmentation model (SAM) and a pretrained semantic feature model (DINOv2) to allow generalization to new objects.

clones the actions from demonstrations $D$, which is optimized with action supervision, a surrogate objective of the respective task success rates.

**Dimensions of Generalization** As detailed in Sec.1, this work examines three dimensions of policy generalization against visual variations: background changes, different camera viewpoints, and new object instances. We outline the scope of generalization considered in this work to avoid ambiguities: 1) *Background changes*: Except for the foreground objects that are involved in a specific manipulation task, any other visual changes in the scene are considered background changes. 2) *Different camera viewpoints*: The cameras during deployment can be mounted at different angles from the training scene. We assume the camera configurations (both extrinsic and intrinsic) are known, as these values can be easily obtained through camera calibration methods [50, 51] in tabletop manipulation environments. 3) *New object instances*: We consider new objects in the same category as those seen in the demonstration trajectories. They differ mainly in color, size, and geometry. Fig. 5 in Appendix visualizes the range of new objects we use for evaluation.

### 3.2 Building Object-Centric 3D Representations

Our main goal is to build representations with both object-centric and 3D-aware properties that allow policies to generalize beyond the training conditions. In the following, we describe the three major steps of building the object-centric 3D representations: 1) scribble annotation to identify task-relevant objects, 2) tracking the objects at the segmentation level to keep models task-focused and exclude the irrelevant visual factors; 3) back-projecting object segmentation to 3D point clouds to be invariant to camera view changes.

**Scribble Annotation of Task-Relevant Objects** In order to build representations that consistently attend to task-relevant objects, the first step is to inform our model of what they are in image observations. Prior work of unsupervised object discovery requires extensive data while being limited to toy domains [46]. Instead, we adopt a semi-supervised approach that annotates objects on a single image with an instance segmentation mask after demonstrations are collected. To operationalize the semi-supervised approach, we leverage an interactive segmentation model, S2M [19], to allow demonstrators to scribble on an image frame with a few mouse clicks, specifying the foreground objects involved in a specific manipulation task, namely the task-relevant objects. More importantly, the annotation only needs to be done on *a single image*. We use a pretrained vision model to track object segmentation in the rest of the datasets, which we describe below. We select the first frame from a demonstration trajectory where task-relevant objects are visible for the demonstrator to provide the scribble annotations. We annotate objects in segmentation, which brings two-fold benefits: First, segmentation outlines objects on the pixel levels, allowing demonstrators to annotate objects with

fine-grained contours. Second, segmentation can effectively exclude irrelevant visual features as opposed to less flexible representations such as bounding boxes, improving the robustness of policies.

**Tracking Task-Relevant Objects Over Time** The first step of scribble annotation indicates task-relevant objects in a single frame. To keep policies focused on these objects while being robust to large visual variations, we need to track them across temporal frames. We employ a pretrained Video Object Segmentation (VOS) model, XMem [17], that takes in the single-frame instance mask from the scribble annotation step and returns a sequence of instance masks that track objects across all the demonstration frames. For all the trajectories in the demonstrations, we can apply XMem to propagate the annotation across trajectories by simply adding the annotated frame at the beginning of the trajectories. In this way, we can annotate the 2D segmentation of task-relevant objects on all images in the datasets. During deployment, we treat the annotated frame as the initial observation and subsequently stream the temporal observations to XMem. In this way, GROOT can segment the objects from image observations and back-project them into point clouds for policy inference.

**Backprojection to Point Clouds** Tracking objects across time frames gives us 2D segmentation masks. However, objects in 2D images are tailored to the specific camera viewpoint; henceforth, they are not robust to camera view changes. To overcome this issue, we need 3D-aware representations that are not specific to camera angles. Concretely, we back-project the segmented objects from 2D images to point clouds using depth images [20]. The point clouds can encode object geometry and allow spatial reasoning while avoiding overfitting to a specific instance as opposed to its RGB-D counterpart. Using point clouds also allows easy generalization to new objects, which we describe later in Sec. 3.4. Then, to represent the point clouds regardless of camera locations, we transform the point clouds into a predetermined reference frame through SE(3) transformation based on the known camera extrinsic [52]. We choose the robot base as the reference frame of coordinates, which is a fixed-base frame of coordinates that does not change with joint configurations or camera setups. Consequently, the transformed point clouds are spatially invariant to changes in camera views, ruling out the potential of significant distribution shifts introduced by viewpoint changes.

### 3.3 Policy Design

In order to harness our object-centric representations, we design a policy architecture, which first encodes the point clouds into a discrete set of tokens and processes them with a transformer-based architecture. GROOT first divides each point cloud into clusters of points in the same procedure as in Point-MAE [53]. It encourages policies to attend to the local geometry of point clouds, and the division is amenable to performing random masking on the inputs. As we show in our ablation study, random masking improves the robustness of policies in terms of large camera changes. Before performing random masking, we pass each point cloud cluster through a shared PointNet [54] to compactly represent object geometry and spatial information in latent vectors, making the subsequent transformer training and inference computationally tractable.

Random masking prevents policies from overfitting to global features of point clouds such that the representations remain robust when partial point clouds are missing due to realistic noise in sensing or large shifts in camera viewpoints. These masked latent vectors, also known as tokens, are passed through a transformer-based architecture that processes object-centric representations [3]. In practice, we pass a $H$-timestep sequence of tokens, allowing the policy to keep track of the temporal information and mitigate the partial observability issue (*e.g.*, a single image does not inform the velocities) and bypassing the non-Markovian property of gripper actions (*e.g.*, a single image cannot differentiate ongoing actions of opening versus closing). Because input to transformers is permutation-invariant, we add sinusoidal positional encoding [21] to the input tokens based on their positions in the temporal sequence (More details in Appendix B).

The action generation of the transformer architecture follows the design of prior work, where an action token is appended to the input sequence, and the transformer outputs the corresponding latent vector for performing downstream tasks. Such designs allow the output latent vector to implicitly encode the object-centric and 3D information from the input sequence through the self-attention mechanisms. The output latent vector is subsequently processed through a Gaussian-Mixture-Moel-

based multi-layer perception (MLP) [1, 55], outputting a distribution of robot action. During training, the network is optimized over a negative log-likelihood objective to mimic the distribution of actions from demonstrations. At test time, the robot samples action from the distribution at every decision-making step, performing closed-loop manipulation at 20 Hz.

### 3.4 Segmentation Correspondence Model

To generalize to new object instances during deployment, we also introduce a segmentation correspondence model that propagates instance segmentation masks to new object instances without additional human annotations.

This model first uses an open-vocabulary segmentation model, SAM [14], to generate a set of object masks from visual observations of deployment environments. Then, the model identifies the object masks that match closest to the ones among the training objects. The closest match is determined by a pretrained semantic feature model, DINOv2 [16]. DINOv2 is a large vision model that is optimized to map similar concepts to close embeddings in its feature space, making it possible to find the closest match by computing the cosine similarity scores between DINOv2 features of objects. In our model, we first compute the DINOv2 feature of every segmentation mask from SAM and every annotated object mask from the training phase. For each pair of SAM segmentation masks and annotated masks, we find the closest matching SAM masks by deciding the mask with the highest cosine similarity score. For every annotated object mask, we select a mask from SAM outputs with the highest similarity score, resulting in a new instance segmentation. This model can, therefore, propagate the object masks from training environments to test scenes with previously unseen objects. Once we have the segmentation of new objects, we can back-project the object masks into point clouds. Consequently, our policies can generalize based on these point clouds due to their similarity with the point clouds of the training objects. As shown in Fig. 1, this correspondence model allows the policy to manipulate different mugs even though it only trains on a single mug during training.

## 4 Experiments

We conduct experiments to answer the following questions: 1) Are object-centric 3D representations in GROOT better at generalization compared to baseline methods? 2) What design choices are essential for good performance of GROOT policies? 3) How well does GROOT generalize in real-world settings in the face of background changes, different camera angles, and new object instances?

### 4.1 Experimental Setup

We conduct quantitative evaluations in physical simulation and the real world. We validate our approach by evaluating policies on three simulation tasks and five real robot tasks, which cover a wide range of contact-rich manipulation behaviors involving prehensile and non-prehensile motions.

**Task Designs**   For simulation experiments, we use three simulation tasks from a lifelong robot learning benchmark, LIBERO [56]. These three simulation tasks are namely "Put the moka pot on the stove", "Put the frying pan on the stove", and "Reposition the yellow and white mug". For real robot experiments, we evaluate five tasks, namely "Pick Place Cup", "Stamp The Paper", "Take The Mug", "Put The Mug On The Coaster", and "Roll The Stamp", which involve behaviors from pick-and-place to non-prehensile motions such as rolling the stamp. We describe how we decide the success conditions of these five tasks in Appendix C.

In our experiments, we have the following testing setups that evaluate the generalization abilities of our policies: 1) `Canonical`: All the objects and sensors are initialized in the same distributions as in demonstrations; 2) Three variants for generalization tests: `Background-Change`, `Camera-Shift`, and `New-Object`. In the simulation, because of the limited availability of object assets, we mainly focus on generalizing different background changes and camera angles. Specifically, we consider two levels of difficulty for each generalization test in simulation, denoted as `Background(Easy)`, `Background(Hard)`, `Camera(Easy)`, and `Camera(Hard)`. Fig. 6 visualizes these variations in the appendix.

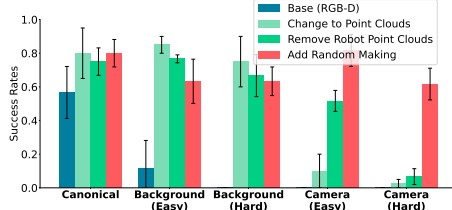
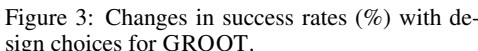

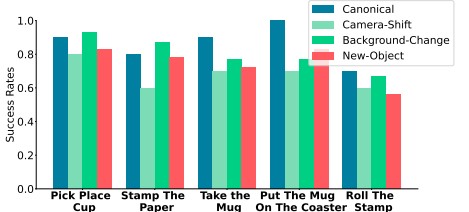

Figure 3: Changes in success rates (%) with design choices for GROOT.

Figure 4: Success rates (%) of GROOT in real-robot tasks.

**Evaluation Metric** We quantify the policy performance by evaluating their task success rates. In the simulation, we evaluate all the models for 20 episodes and average the results over models trained with three random seeds. In real-world settings, for `Canonical`, and `Camera-Shift`, we evaluate 10 episodes for each task. For `Background-Change`, we evaluate 30 episodes that cover a diverse range of table colors, lighting conditions, and distracting objects. For `New-Object`, we evaluate new objects three times. Fig. 5 visualizes the new objects we use for evaluation.

| Tasks | Evaluation Setting | BC-RNN | VIOLA | MAE | GROOT |
|---|---|---|---|---|---|
| Put the moka pot on the stove | `Canonical` | $56.7 \pm 15.5$ | $61.7 \pm 27.2$ | $73.3 \pm 6.2$ | **80.0**$\pm 8.2$ |
| | `Background(Easy)` | $11.7 \pm 16.5$ | **77.5**$\pm 17.5$ | $56.7 \pm 29.5$ | $63.3 \pm 13.1$ |
| | `Background(Hard)` | $0.0 \pm 0.0$ | $0.0 \pm 0.0$ | $31.7 \pm 16.5$ | **63.3**$\pm 8.5$ |
| | `Camera(Easy)` | $0.0 \pm 0.0$ | $0.0 \pm 0.0$ | $45.0 \pm 20.4$ | **81.7**$\pm 9.4$ |
| | `Camera(Hard)` | $0.0 \pm 0.0$ | $0.0 \pm 0.0$ | $26.7 \pm 9.4$ | **61.7**$\pm 9.4$ |
| Put the frying pan on the stove | `Canonical` | $51.7 \pm 6.2$ | **90.0**$\pm 4.1$ | $81.7 \pm 4.7$ | $65.0 \pm 10.8$ |
| | `Background(Easy)` | $10.0 \pm 10.8$ | $62.5 \pm 2.5$ | **73.3**$\pm 16.5$ | $61.7 \pm 11.8$ |
| | `Background(Hard)` | $0.0 \pm 0.0$ | $6.7 \pm 4.7$ | $52.5 \pm 7.5$ | **58.3**$\pm 11.8$ |
| | `Camera(Easy)` | $0.0 \pm 0.0$ | $1.7 \pm 2.4$ | **76.7**$\pm 20.5$ | $71.7 \pm 12.5$ |
| | `Camera(Hard)` | $0.0 \pm 0.0$ | $1.7 \pm 2.4$ | $65.0 \pm 7.1$ | **73.3**$\pm 8.5$ |
| Reposition the yellow and white mug | `Canonical` | $63.3 \pm 16.5$ | **85.0**$\pm 8.2$ | $66.7 \pm 8.5$ | $73.3 \pm 2.4$ |
| | `Background(Easy)` | $53.3 \pm 30.9$ | $81.7 \pm 6.2$ | $38.3 \pm 12.5$ | **81.7**$\pm 12.5$ |
| | `Background(Hard)` | $0.0 \pm 0.0$ | $48.3 \pm 31.7$ | $6.7 \pm 9.4$ | **83.3**$\pm 6.2$ |
| | `Camera(Easy)` | $0.0 \pm 0.0$ | $3.3 \pm 4.7$ | $55.0 \pm 16.3$ | **61.7**$\pm 18.9$ |
| | `Camera(Hard)` | $0.0 \pm 0.0$ | $5.0 \pm 4.1$ | $8.3 \pm 8.5$ | **38.3**$\pm 12.5$ |

Table 1: We conducted a quantitative evaluation in simulation to investigate policy generalization. The table reports the success rates (%) averaged over three random seeds.

## 4.2 Experimental Results

We answer question (1) by comparing GROOT policies in the three simulation tasks against three baselines that also learn closed-loop visuomotor policies: BC-RNN [1, 8] that directly conditions on raw perception, VIOLA [3] that uses object-centric priors, and MAE-POLICY [57] that uses random masking on image patches for learning policies. BC-RNN is a baseline to present the lowest bound of performance we can achieve in the canonical settings of the presented simulation tasks. More details are in Appendix B for baseline implementations.

Table 1 presents the full simulation results that show the effectiveness of our representations in learning robust policies. As the table shows, while previous approach such as VIOLA excels at the canonical settings in some tasks, GROOT significantly outperforms baseline methods in most of the generalization tests by an amount of 22% success rates compared to the best baseline, MAE-POLICY. The comparison between VIOLA and GROOT suggests that while task-agnostic object priors are good at handling easy background changes, such priors significantly fail when the background changes are more adversarial, as in the `Background(Hard)` case. This result supports the choice of our representations that directly attend to task-relevant objects, excluding task-irrelevant visual factors. The comparison between GROOT and MAE-POLICY policies shows that random masking generally improves the robustness of policy learning across different variations. However, naively dividing observations into patches impedes the policies from having decent performance in hard levels of generalization tests. This comparison further supports our policy learning centered around objects, which consistently give us a good performance on hard levels of generalization tests.

Furthermore, in some simulation tasks and real-robot tasks, policies are better in some generalization tests than in canonical setups. We posit that it results from the efficacy of the object-centric 3D representations in GROOT, where visual variations in image observations cease to be the only limiting factor for policy generalization. In particular, policies trained over the supervised learning objective of BC do not necessarily achieve the best task success rates, and the BC objective is only a surrogate function of maximizing task success rates. As a result, our policies perform better in various generalization tests against visual variations. As the primary focus of GROOT is to develop visuomotor policies that generalize to visual variations beyond the one in training conditions, we leave for future work to improve other aspects of the visuomotor policy learning.

**Important Design Choices** Here, we present an ablation study on the task "Put the moka pot on the stove" to answer the second question, showing how our model takes advantage of the representations. Fig. 3 visualizes the change in performance in using point clouds, removing point clouds of the robots, and using random masking on point clouds for training policies. We start with the base model that naively takes in RGB-D observations. This base model fails to generalize in all four test settings. To mitigate irrelevant visual factors, we adopt an ablative model that inputs only point clouds of objects and the robot, enhancing generalization for different background changes and validating the effectiveness of our object-centric priors. Despite these improvements, the model struggles with camera view changes due to the inclusion of unseen robot parts in the observations. To address these distribution shifts, we exclude the robot point clouds from observations, leveraging known robot models *a priori*. This adjustment enhances performance under minor camera view changes, yet struggles with substantial view alterations. To accommodate large view changes, we incorporate random masking into the point cloud representation as detailed in Sec. 3. Consequently, we achieve a substantial performance boost in handling camera view changes, albeit at a modest reduction in background change generalization capability.

**Real Robot Experiments** We answer the third question by presenting results in Fig. 4. The table shows that our policies successfully generalize to all three variations with high success rates despite the challenging settings. Our real-robot experiments validate the effectiveness of our approach in the experiments. We include all the videos of evaluation rollouts on our project website. Despite the great performance of GROOT, the policies still fail to achieve perfect success rates. Most of the failures come from missing the grasp by a few millimeters. This is within our expectation as the workspace cameras do not have enough resolution to take into account this level of location errors, and we do not include the eye-in-hand cameras in our real robot experiments. Eye-in-hand cameras have been proven effective in learning robust grasping [58]. We expect a performance boost by adding the eye-in-hand camera for precise grasping, but this hardware setup is orthogonal to our research topic, and we leave this for future work.

## 5 Conclusion

We present GROOT, an imitation learning method for learning closed-loop manipulation policies based on object-centric 3D representations. Our approach leverages object segmentation and point clouds to build effective representations that allow the policy to generalize to background changes, different camera angles, and new object instances. Our results show that GROOT outperforms state-of-the-art baselines in terms of its generalization abilities, and we validate our methods in five real robot tasks, showcasing its superior generalization abilities in settings beyond its training conditions.

**Limitations** In this work, we assume only one instance of each task-relevant object is present in a task. When multiple instances of the same object category are present, the segmentation correspondence model would consider all of them as candidate objects. In the future, we will consider language-conditioned policies with a spatial specification to resolve the ambiguities.

Our approach focuses on achieving wide generalization across perceptual variations, but assuming the robot morphology must be the same. Extending to generalization across different morphologies will be an interesting extension (e.g., between different tabletop manipulators, or even from tabletop to mobile manipulators). Such knowledge transfer would require a unified design of action space, which we leave for future work.

**Acknowledgements**  The authors would like to thank Vincent Cho, Jeffery Zhang, and Jake Grigsby for the insightful discussion on the project and the manuscript. The authors would also like to thank Yuzhe Qin for the helpful discussion on the depth cameras. This work has taken place in the Robot Perception and Learning Group (RPL) and Learning Agents Research Group (LARG) at UT Austin. RPL research has been partially supported by the National Science Foundation (CNS-1955523, FRR-2145283), the Office of Naval Research (N00014-22-1-2204), and the Amazon Research Awards. LARG research is supported in part by NSF (FAIN-2019844, NRT-2125858), ONR (N00014-18-2243), ARO (E2061621), Bosch, Lockheed Martin, and UT Austin's Good Systems grand challenge. Peter Stone serves as the Executive Director of Sony AI America and receives financial compensation for this work. The terms of this arrangement have been reviewed and approved by the University of Texas at Austin in accordance with its policy on objectivity in research.

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

# A  Experiment Details

## A.1  New Object Generalization

In Fig. 5, we show pictures of seen / unseen objects. For all the new object generalizations, we evaluate policies on multiple objects despite each policy being trained on a single object. Across tasks, we showcase the new objects that involve variations in color and geometry.

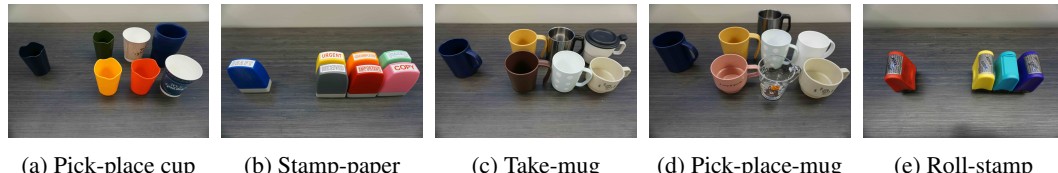

(a) Pick-place cup     (b) Stamp-paper     (c) Take-mug     (d) Pick-place-mug     (e) Roll-stamp

Figure 5: Overview of objects used in real-robot experiments. In each image, the single object on the **left** side is used during data collection, and **all** the objects on the right side are not seen during training. They are used for the evaluation of new object generalization in each task.

# B  Additional Implementation Details

We describe all the details of our model implementation aside from the ones mentioned in the main text.

## B.1  Model Details

**Neural Network Details.** We use a standard Transformer [21] architecture in our paper. We use 4 layers of transformer encoder layers, and 6 heads of the multi-head self-attention modules. For the two-layered fully connected networks, we use 1024 hidden units for each layer. For GMM output head, we choose the number of modes for the Gaussian Mixture Model to be 5, which is the same as in Mandlekar et al. [1].

Additionally, we summarize in Table 2 what components are pre-trained and frozen, and what components are trained from scratch.

| Components | Pre-trained or From Scratch |
|---|---|
| S2M [19] | Pre-trained |
| VOS [17] | Pre-trained |
| DinoV2 [16] | Pre-trained |
| SAM [14] | Pre-trained |
| PointNet++ [54] | From Scratch |
| Transformer [21] | From Scratch |
| GMM-MLP [1, 55] | From Scratch |

Table 2: This table presents which component is pre-trained and then frozen in our experiments, and which is trained from scratch. We denote them as Pre-trained and From Scratch, respectively.

**Temporal Positional Encoding.** For computing temporal positional encoding, we follow the equation for each dimension $i$ in the encoding vector at a temporal position $pos$:

$$PE(pos, 2i) = \sin\left(\frac{pos}{10^{2i/D}}\right)$$
$$PE(pos, 2i + 1) = \cos\left(\frac{pos}{10^{(2i+1)/D}}\right)$$

We choose the frequency of positional encoding to be 10 which is different from the one in the original transformer paper. This is because our input sequence is much shorter than those in natural

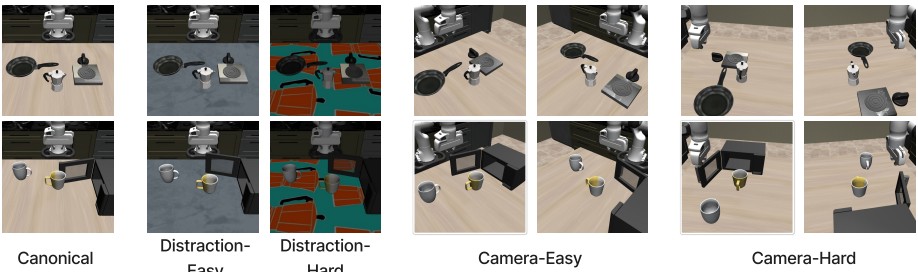

| Canonical | Distraction-Easy | Distraction-Hard | Camera-Easy | Camera-Hard |

Figure 6: Screenshots of simulation tasks, for both `Canonical`, `Background(Easy)`, `Background(Hard)`,`Camera(Easy)`,`Camera(Hard)`

language tasks, hence we choose a smaller value to have sufficiently distinguishable positional features for input tokens. Note that all the input tokens at the same timestep are added with the same positional encoding to inform the transformer about the temporal order of received.

**Number of Clusters in Point-MAE**    Across all experiments, we choose 10 for the number of clusters in Point-MAE.

Here, we also denote the dimensions of inputs and outputs in the process of Point-MAE. Suppose that we have a 3D point cloud that consists of $P$ points. The input to Point-MAE is a vector in $\mathbb{R}^{P \times 3}$. Point-MAE groups the point cloud into $c$ clusters, with $p$ points in each cluster. This operation gives us grouped point clouds that are in $\mathbb{R}^{c \times p \times 3}$. The grouped point clouds are passed through a PointNet++ encoder, giving us latent vectors in $\mathbb{R}^{c \times p \times h}$, where h is the dimension of latent features. Finally, given a masking ratio $m$, the actual latent vectors that are included in the transformer input are vectors in $\mathbb{R}^{(1-m)c \times p \times h}$.

**Training Details.**    For point masking, we use a masking ratio of $0.6$ in simulation, and $0.75$ for real world. Because of the limited field of view, the occlusion of objects in simulation is severe. To properly evaluate policies in simulation, we add an eye-in-hand camera that only captures close-distance depth (the depth observation is clipped to the range of gripper tips). This design choice allows the policies to learn while preventing policies from relying entirely on eye-in-hand cameras.

In all our experiments of GROOT, we train for 100 epochs. We use a batch size of 16 and a learning rate of $10^{-4}$. We use negative log-likelihood as the loss function for action supervision loss since we use a GMM output head. As we notice that validation loss doesn't correlate with policy performance [1], we adopt a pragmatic way of saving model checkpoint as in Zhu et al. [3], which is to save the checkpoint that has the lowest loss over all the demonstration data at the end of training. We apply a gradient clip at 100 across all the experiments to prevent training from gradient explosion.

## C   Environment Details

Here we describe more about our environment designs.

**Baseline Implementations.**    As we mainly focus on comparing the effectiveness of representations, all the transformer-based baselines (VIOLA and MAE-POLICY) use the same architecture for a fair comparison. Since none of the baselines were proposed for learning with RGB-D observations, we implemented them with minimal changes to accommodate the RGB-D observations. For BC-RNN, we encode depth images with an additional resnet encoder, and concatenate the features along with the other features as inputs to the RNN backbone. For VIOLA, we extract the task-agnostic proposals and back-project each proposal into point clouds, giving VIOLA a fair

comparison with our approach. As for MAE-POLICY, we patchify both RGB and depth images and pass the unmasked patches into the transformer architecture.

**Generalization Settings in Simulation.** Fig. 6 shows the initial conditions of simulation tasks. Note that "Put the moka pot on the stove" and "Put the frying pan on the stove" share the same initial distributions, so we only visualize one of the tasks for showing the initialization settings. `Background(Hard)` is the hard level as we changed both the lighting conditions, and added the table cloth that has object patterns. `Camera(Hard)` is harder than `Camera(Easy)`, as the cameras are rotated with 40 more degrees. Such a wild change in camera viewpoints results in a very different perspective on objects. Challenges the generalization abilities of policies.

**Real-Robot Setup.** We use a 7-DoF Franka Emika Panda arm in all tasks. For real robot end-effector control, we use the Operational Space Controller [59] implemented from Deoxys [3]. The controller operates at 20Hz alongside a binary gripper control. We use Intel Realsense D435i as the workspace camera.

**Success Conditions of Real-Robot Task.** To quantify the policy performance, we explain the success conditions for all the tasks as follows:

- "Pick Place Cup": The cup is placed on the coaster upright.
- "Stamp The Paper": The robot stamps on the paper and put the stamp back to the table
- "Take The Mug": The mug is taken from the coaster, and placed on the table steadily.
- "Put the Mug On The Coaster": The mug is put on the coaster steadily.
- "Roll the Stamp": The robot successfully rolls the stamp for half of the paper length.

**Data Collection.** We use a 3Dconnexion SpaceMouse to collect 50 human-teleoperated demonstrations for every real-world task. As for simulation, the simulation environments directly provide 50 high-quality teleoperated demonstrations, so we directly leverage them for policy learning.

**Evaluation Horizons.** For policy evaluation, we limit the decision horizons to 600 timesteps.

**Initial Conditions.** To systematically evaluate policies under such conditions, we pre-specify a region and randomly place the objects inside the region. This region is used for object placements during demonstration collection as well as policy evaluation. Here, 7 shows an overlay image of initial frames from three representative rollouts of the task "Put The Mug On The Coaster" during camera generalization tests. As the figure shows, the mug is placed at different places with different orientations at the beginning of rollouts. The objects will be placed at locations that do not overlap with other locations. In all, the initial conditions are randomized such that policies really need to go toward the correct location of an object in order to achieve high success rates.

## D  Additional Ablation Study

### D.1  Point Cloud Encoders

Here we provide an ablation study to study the importance of point net encoders in GROOT. We use the simulation task "Put the moka pot on the stove" to compare the PointNet++ architecture used with two other common architectures for processing point clouds, namely DGCNN [60] and Point Transformer [61]. We show the ablation study in Table 3. We see that Point Transformer-based encoder generally performs better than the DGCNN-based one, but neither of them brings significant improvements to the policy. We include this ablation study for ease of future reference.

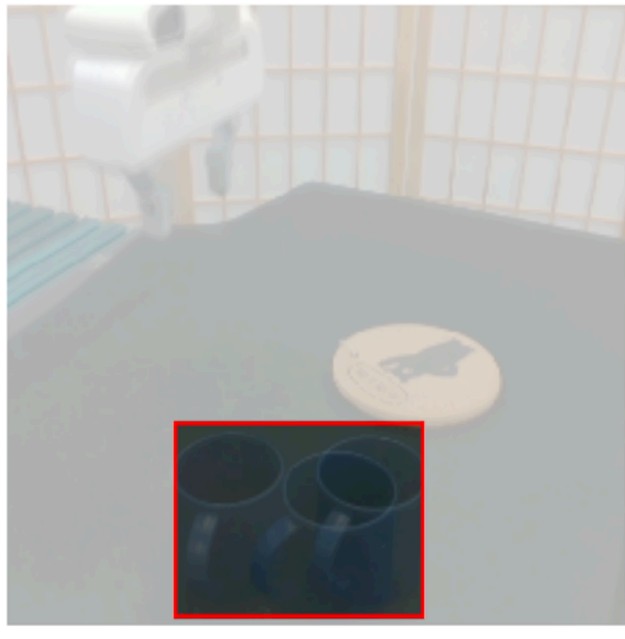

Figure 7: Overlay image of initial frames from three representative rollouts of the task "Put The Mug On The Coaster" during camera generalization tests.

| Point cloud encoder | PointNet++ | DGCNN (GNN-based) | Point Transformer |
|---|---|---|---|
| Canonical | $80.0 \pm 8.2$ | $65.0 \pm 4.1$ | $67.5 \pm 7.5$ |
| Background(Easy) | $63.3 \pm 13.1$ | $63.3 \pm 6.2$ | $77.5 \pm 7.5$ |
| Background(Hard) | $63.3 \pm 8.5$ | $63.3 \pm 4.7$ | $75.0 \pm 5.0$ |
| Camera(Easy) | $81.7 \pm 9.4$ | $65.0 \pm 0.0$ | $60.0 \pm 0.0$ |
| Camera(Hard) | $61.7 \pm 9.4$ | $63.3 \pm 8.5$ | $75.0 \pm 5.0$ |

Table 3: Ablation study on the choice of point cloud encoders.

