# OpenReview forum: "Learning Generalizable Manipulation Policies with Object-Centric 3D Representations"
_robot-learning.org/CoRL/2023/Conference — CoRL 2023 Poster_

### Official Review · Reviewer_ghms · 2023-06-24

**Confidence:** 4
**Originality:** Very Good
**Technical Quality:** Excellent
**Clarity Of Presentation:** Very Good
**Impact:** 4

**Recommendation:**

Strong Accept: I recommend accepting the paper and will argue for my recommendation even if other reviewers hold a different opinion.

**Review:**

Strengths:
* \+ real robot experiments.
* \+ strong performance, significant gains over baselines.
* \+ generalization capabilities; though it requires another pre-trained representation model at inference time (DINO).
* \+ converting RGB-D to PC is an interesting approach to allow generalization in manipulation tasks, avoiding distracting signals in images such as colors and brightness/contrast.

Weaknesses/Limitations:
* \- while the user interaction is rather minimal, I find the requirement to annotate/segment by the user for each scene a limitation of this method.
* \- The approach proposed in this paper employs several pre-trained modules such as S2M, SAM, DINO and XMem. While I don’t find this a significant limitation, I suggest that for better clarity the authors will organize the pre-trained and training models in a table (or something else) to help the reader understand which parts of the model are trained and which are frozen. For example, a table with columns: Model (e.g., S2M, DINO), Purpose (e.g., Segmentation, Tracking), Pre-trained/Trained, etc…

Minor Comments:
* Lines 258-259: “; 2) Three variants for generalization tests: , Camera-Shift, and New-Object. “, I believe the “Background-Change” variant is missing.
* Line 280: missing word after “and”: “The comparison between VIOLA and [missing] suggests”.


**Quality Of The Limitations Section:**

Limitations are addressed clearly

**Questions For Rebuttal:**

* Backprojection to PCs: is it done via an external library (Open3D) or something else? I see you cite it but it is not clear what is the input and output of that step.
* Transformer input (lines 189-198): if I understand correctly, each object is represented as a PC, which is then clustered to groups. Each group is fed through PointNet++ with global pooling to output a single vector which is then finally used as input to the transformer. It would be beneficial to clarify this step with mathematical notations, i.e., the inputs and outputs dimensions. Is the number of clusters (e.g., 3) per PC is predetermined (a hyper-parameter)?
* Positional encoding in the transformer: at timestep $t$, tokens (=clusters) of the same object get the same temporal encoding? Similarly, different objects but both at timestep $t$ get the same temporal encoding? Several object-centric works considered different positional embeddings to better capture spatio-temporal relationships between objects ([1]-[3]).
* Comparison with VIOLA: in essence, GROOT replaces the RPN for object proposals with user-interaction and S2M to select objects and utilizes depth to backproject to point-clouds. GROOT also inherits the idea of masking tokens. VIOLA also mentions that they use multiple views to handle occlusions. While the authors mentioned that for a fair comparison they modified VIOLA to support depth input (RGB-D), have you tried using VIOLA with multi-view input as a baseline? Alternatively, instead of user-interaction, have you considered using RPN similarly to VIOLA? I believe that a hybrid model of VIOLA and GROOT can lead to another improvement (though I don’t expect the authors to implement it).
* To facilitate further research, are you planning to make the code available?

[1] https://arxiv.org/abs/2107.09240

[2] https://arxiv.org/abs/2210.05861

[3] https://arxiv.org/abs/2306.05957


**Robotics Focus:**

Sufficient demonstration on hardware

**Summary Of Paper:**

This paper proposes GROOT, an imitation learning approach for visual-based manipulation by learning structured 3D object-centric representations. They utilize a transformer-based policy over point-clouds, structured to utilize the object-centric inductive bias, leading to performance gains in manipulation tasks, both in simulated and real environments. At inference time, to allow generalization, they use a general segmentation module (SAM) to segment the entire observation, and utilize a pre-trained representation module (DINO) to find the objects-of-interest via cosine-similarity and propagate them to the trained network.

**Summary Of Recommendation:**

Overall, I like the idea of this paper and I’m convinced that this approach can help generalization. In addition, simulation and real robot experiments show that indeed the object-centric approach is crucial. Given my questions are addressed, I would like to recommend this paper for acceptance.

---

### Official Review · Reviewer_p4GH · 2023-06-29

**Confidence:** 4
**Originality:** Good
**Technical Quality:** Very Good
**Clarity Of Presentation:** Excellent
**Impact:** 3

**Recommendation:**

Weak Accept: I recommend accepting the paper, but will not argue for my recommendation if the majority of other reviewers have a different opinion.

**Review:**

The paper introduces a novel approach, GROOT, which combines object-centric and 3D priors to address the challenges of generalization in perception for manipulation tasks. This approach takes advantage of structured visual representations to enhance the robustness and adaptability of visuomotor policies. GROOT uses state-of-the-art visual recognition models and interactive segmentation techniques to construct object-centric 3D representations. It leverages depth information and back-projects segmentation masks into 3D point clouds, capturing object information and spatial relations in 3D space. The paper provides a comprehensive evaluation of GROOT in both simulated and real-world environments. The results show that GROOT outperforms baseline methods in terms of generalization abilities, handling background changes, different camera angles, and new objects. The training process, however, is rather manual, which requires manual labor of segmenting the images to annotate the training data. Furthermore, the paper lacks an intuitive explanation of why the method performs better than baselines such as VIOLA, as most of the components in the system are "standard".


**Quality Of The Limitations Section:**

Limitations are addressed clearly

**Questions For Rebuttal:**

1. Have the authors experimented wit other point cloud encoders?
2. Is it possible to reduce human input during training data generation?
3. Would it be possible to explain intuitively why GROOT performs better at generalization?

**Robotics Focus:**

Highly relevant to robotics but no hardware experiments

**Summary Of Paper:**

The paper proposes a method called GROOT for learning generalizable manipulation policies using object-centric and 3D representations. The main contributions of the paper include the construction of object-centric 3D representations robust to background changes and camera views, the use of a transformer-based policy for reasoning over the representations, and the introduction of a segmentation correspondence model for reusing strategies in the case of new object manipulation. The paper presents comprehensive experiments in simulated and real-world environments to validate the effectiveness and robustness of the proposed method.

**Summary Of Recommendation:**

Very solid experiments section but some ideas are not expressed clearly due to the lack of explanation.

---

### Official Review · Reviewer_Yp2B · 2023-07-17

**Confidence:** 4
**Originality:** Good
**Technical Quality:** Very Good
**Clarity Of Presentation:** Good
**Impact:** 3

**Recommendation:**

Weak Accept: I recommend accepting the paper, but will not argue for my recommendation if the majority of other reviewers have a different opinion.

**Review:**

While I think that the performance of this model is pretty good relative to VIOLA, I think this fact can be attributed to the better vision models used here. I do not see very many novel ideas in this paper. The policy backbone (a big transformer with a standard configuration) seems similar to many other works. The preprocessing on the input seems to be a basic application of the vision models, i.e. object detection, segmentation, dino2, etc.

In terms of results, the model performs pretty well, outperforming VIOLA in situations where there is a significant shift in background or camera perspective.

In summary, I think this is notable work, but the lack of novelty dampens my enthusiasm. It would have been good to expand table 1 to include results from a larger number of tasks.

**Quality Of The Limitations Section:**

Limitations are addressed clearly

**Questions For Rebuttal:**

I didn't see anywhere in the paper where you mention the number of demonstrations used in your experiments. That's information you should probably include.

**Robotics Focus:**

Sufficient demonstration on hardware

**Summary Of Paper:**


The paper introduces GROOT, an approach to imitation learning that uses a transformer backbone in combination with object embeddings obtained from standard large vision models. Roughly, there are two stages: vision preprocessing and then IL using the transformer.

In the preprocessing step, the human user provides demonstrations wherein the task-relevant objects are identified using a scribbling interface. These objects are tracked through the rest of the frames of the demonstration using a tracking model. At time time, the model attempts to find the same objects as from the demonstrations using dino2 features.

In the IL step, the model uses a transformer where the tokens are 3d models of the objects obtained by projecting the test-time object segments onto the point cloud. The point cloud object representations are clustered and tokenized. A standard policy transformer is used where a specific action token is introduced to generate the action.


**Summary Of Recommendation:**

I didn't see many novel ideas here. Nevertheless, the method outperforms VIOLA on three relevant tasks, so there is a contribution.

---

### Official Review · Reviewer_YKj9 · 2023-07-22

**Confidence:** 4
**Originality:** Fair
**Technical Quality:** Fair
**Clarity Of Presentation:** Fair
**Impact:** 2

**Recommendation:**

Weak Accept: I recommend accepting the paper, but will not argue for my recommendation if the majority of other reviewers have a different opinion.

**Review:**

Quality: This paper addresses an important generalization challenge in robot learning, providing a reasonable algorithm that bakes in object and camera-viewpoint into the method. While the methodology is reasonable, some experimental results and evaluation setups can be under question.

Clarity: The paper is well-structured, however there are many typos and missing words, in addition to some technically incorrect statements.

Strengths:
1. The paper targets a major challenge in robotics which is generalization to visual scene perturbations and camera viewpoint changes. The multi-step methodology for segmenting out objects of interest and conditioning the policy using point clouds always transformed to the same world frame is an interesting method for robotic manipulation which can generalize to camera viewpoint changes.

Weaknesses:
1. Line 126: The authors mention that their “objective is to find a visuomotor policy $\pi$ that maximizes the expected task success rates.” This is incorrect, as for the behavior cloning policy proposed in this paper the objective is to mimic the expert and there are no rewards obtained from the environment that the agent can maximize.
2. Line 257: missing keyword for first generalization test
3. The method needs access to ground truth camera extrinsics in order to transform the observed point clouds to a canonical world frame. This might not always be easy to obtain in real world setups.
4. In the video demos, the initial conditions seem to be very insignificantly perturbed. For camera-viewpoint generalization, while the model is shown to work with observations from varying camera positions, all evaluation rollouts act on almost the same initial setup which could have been cherry-picked.

**Quality Of The Limitations Section:**

Additional details required

**Questions For Rebuttal:**

1. Table 1 seems to show that for all tasks, GROOT had a better success rate with some environmental perturbations than in the canonical setup. Same case for success rates on two real-robot tasks, where the proposed method performed worse in the canonical setup than when background was changed. How was that possible?
2. One clear limitation of the work to me is the need for camera extrinsics for transforming input point clouds to canonical world frame. The need for camera-viewpoint generalization in learned policies is usually the lack of such transformations being readily available, and the expectation from the model to learn them implicitly. Can the authors discuss this limitation in detail and its implications when deploying their method on a real robot?

**Robotics Focus:**

Sufficient demonstration on hardware

**Summary Of Paper:**

This paper proposes an imitation learning method aimed at learning robust policies that generalize beyond their training configurations. They construct an object-centric 3D representation by segmenting out objects in a point cloud and using them to condition an imitation learning policy. The authors argue that incorporating structural visual representations into the design of an imitation learning method is key to handling visual variations in the real world, and propose a method for transforming input point clouds to a canonical world frame to tackle varying camera viewpoint changes. The authors show experiments on tasks in simulation as well as the real world, as well as perform generalization experiments to show invariance against background changes, camera viewpoints, and new object instances.

**Summary Of Recommendation:**

The paper needs a major revision as there are multiple missing words, typos, grammatical errors, as well as some technically incorrect statements. There are also some missing discussions about limitations of the proposed method in my opinion, which also bring the impact of this work on real-world robot deployment into question. Additionally, the experimental results (doing worse on canonical than when perturbed) make it seem that there is potentially a performance gap between the current state of the method and what it’s ideal implementation would be.

---

> ### Author Response · Authors · 2023-08-15
> **Follow-up Response to Reviewer YKj9**
>
> Dear Reviewer YKj9,
>
> We would like to thank the reviewer again for their efforts and time in providing thoughtful feedback and comments. We’ve revised the paper according to the reviewer’s suggestions and replied to all the questions and concerns. If the reviewer has further concerns regarding our work, we welcome any follow-up discussions. If the reviewer has no further concerns,  we sincerely ask the reviewer to consider raising the score. Thank you very much!

---

### Author Response · Authors · 2023-08-16
**Response to all reviewers**

Dear reviewers,

We would like to thank all reviewers again for their valuable input and time, which has enhanced the quality of our paper. We have revised the manuscript and incorporated additional ablation experiments to address concerns raised about our model design choices and experimental results. Specifically, we would like to highlight the following points that are addressed in our replies to reviewers' comments:

* We have clarified that using camera extrinsics is standard practice in robotic systems, and these can be readily obtained.
* We have provided insights about the superior generalization performance of our approach compared to baseline methodologies. Furthermore, we have explained how our policies can sometimes perform better in generalization tests than in canonical setups.
* We have clarified the novelties of our approach, emphasizing its synergetic integration of vision foundation models and a segmentation correspondence model, which unlocks the new object generalization.
* We have highlighted the significance of user interaction for annotating task-relevant objects and provided explanations on how to further enhance this component or replace it in future work.
* We have detailed the initial conditions of our evaluations, demonstrating the object placement variations in our experiments to ensure our results are not selectively showcased.
* We have revised the manuscript, which fixed the mentioned typos, grammatical issues, and some missing words.

We believe that our responses adequately address the main concerns of the reviewers. We are happy to answer any future inquiries. If there are no additional questions, we kindly request a re-evaluation of our paper. We look forward to constructive discussions about our research.

---

### Decision · Program_Chairs · 2023-08-30

**Decision:**

Accept (Poster)

**Comment:**

The authors present an imitation learning method for visual-based manipulation. They use object-centric 3D representations (segmenting out objects in a point cloud) that encode invariance to background appearance and camera position in order to improve robustness of a downstream imitation learning policy. They call the algorithm GROOT and they demonstrate compelling results on both simulated and real-world robotic environments, including ablations to measure generalization and invariance to various distractor features (i.e. background), latent variable variations (e.g. camera pose) and to new object instances.

Reviewers highlighted the importance of generalization for imitation learning as a problem, likely as a result of the author’s clear and concise motivation of the problem itself. (aside: not raised by reviewers, but this AC does wonder if such generalization issues are a result of the small scale of the paper’s datasets, which are mitigated by the recent trend to large internet-scale cotraining). Several clarification issues were raised during the rebuttal process and the authors provided detailed explanations for missing information and additional experiments where appropriate. One reviewer raised their score as a result. The resultant recommendation from all reviewers was towards acceptance.

One reviewer highlighted that the choice to have users annotate and segment each scene is a limitation of the method; a concern that was reiterated at the end of the rebuttal process (after author rebuttals). With that said, this seems like something that could be automated for future work and seems very addressable with some effort - as highlighted in the author’s response (aside: the author’s argument that this user-interaction step increases transparency and user trust would need to be verified).